# ATP13A2 Declines Zinc-Induced Accumulation of α-Synuclein in a Parkinson’s Disease Model

**DOI:** 10.3390/ijms23148035

**Published:** 2022-07-21

**Authors:** Huiling Gao, Hehong Sun, Nan Yan, Pu Zhao, He Xu, Wei Zheng, Xiaoyu Zhang, Tao Wang, Chuang Guo, Manli Zhong

**Affiliations:** 1College of Life and Health Sciences, Northeastern University, Shenyang 110169, China; gaohuiling@mail.neu.edu.cn (H.G.); shh042014@163.com (H.S.); zhaopu6687700@mail.neu.edu.cn (P.Z.); wt7294@hotmail.com (T.W.); guoc@mail.neu.edu.cn (C.G.); 2School of Medical Applied Technology, Shenyang Medical College, Shenyang 110034, China; yannan@symc.edu.cn; 3Department of Anatomy, Histology and Embryology, School of Medicine, Shenzhen University, Shenzhen 518060, China; oliviaxu@szu.edu.cn; 4Department of Histology and Embryology, School of Basic Medical Sciences, China Medical University, Shenyang 110122, China; wzheng@cmu.edu.cn; 5Division of Biotechnology, Dalian Institute of Chemical Physics, Chinese Academy of Sciences, Dalian 116023, China; zhangxiaoyu@dicp.ac.cn

**Keywords:** Parkinson’s disease, α-synuclein, ATP13A2, zinc, lysosome, apoptosis

## Abstract

Parkinson’s disease (PD) is characterized by the presence of Lewy bodies caused by α-synuclein. The imbalance of zinc homeostasis is a major cause of PD, promoting α-synuclein accumulation. ATP13A2, a transporter found in acidic vesicles, plays an important role in Zn^2+^ homeostasis and is highly expressed in Lewy bodies in PD-surviving neurons. ATP13A2 is involved in the transport of zinc ions in lysosomes and exosomes and inhibits the aggregation of α-synuclein. However, the potential mechanism underlying the regulation of zinc homeostasis and α-synuclein accumulation by ATP13A2 remains unexplored. We used α-synuclein-GFP transgenic mice and HEK293 α-synuclein-DsRed cell line as models. The spatial exploration behavior of mice was significantly reduced, and phosphorylation levels of α-synuclein increased upon high Zn^2+^ treatment. High Zn^2+^ also inhibited the autophagy pathway by reducing LAMP2a levels and changing the expression of LC3 and P62, by reducing mitochondrial membrane potential and increasing the expression of cytochrom C, and by activating the ERK/P38 apoptosis signaling pathway, ultimately leading to increased caspase 3 levels. These protein changes were reversed after ATP13A2 overexpression, whereas ATP13A2 knockout exacerbated α-synuclein phosphorylation levels. These results suggest that ATP13A2 may have a protective effect on Zn^2+^-induced abnormal aggregation of α-synuclein, lysosomal dysfunction, and apoptosis.

## 1. Introduction

Parkinson’s disease (PD) is a common neurodegenerative movement disorder, characterized by the progressive loss of dopaminergic neurons in the substantia nigra pars compacta and the presence of intracellular protein inclusions, known as Lewy bodies, in surviving neurons. PD onset is associated with excessive apoptosis in dopaminergic neurons, which leads to the inhibition of the direct dopaminergic pathway, ultimately leading to the loss of motor function and other clinical symptoms [1,2]. α-synuclein is the main component of Lewy bodies. As a 140-amino-acid-long protein, alpha-synuclein is mainly localized at the presynaptic nerve terminal [3,4] and nucleus [5,6,7,8,9]. Under normal physiological conditions, α-synuclein plays an important role in the transport of vesicles [10] and participates in the release, storage, and circulation of neurotransmitters in neurons [11,12]. In addition, α-synuclein is associated with the regulation of intracellular enzymes and facilitates the cellular transport of dopamine [12,13]. Alterations in the copy number of the gene encoding α-synuclein (*SNCA*), such as its duplications and triplications, as well as mutations in this gene (such as A53T, A30P, E64K, and the newly discovered H50Q and G51D) cause autosomal dominant PD [14,15,16,17,18,19]. In subjects who carry additional copies of *SNCA*, the degree of α-synuclein overexpression is directly proportional to the severity of the disease [20,21]. Additionally, mutations in *SNCA* are also important risk factors for sporadic PD, highlighting that α-synuclein plays a key role in PD pathogenesis. Phosphorylation of α-synuclein at serine-129 is a biomarker for pathological forms of α-synuclein aggregation [22,23]. In vivo, α-synuclein exists in the form of monomers, oligomers, fibrils, and aggregates. These different forms of α-synuclein exist in a dynamic equilibrium that is influenced by external (such as pH changes, environmental poisons, and metal ions) and internal factors (such as cytochrome C and apolipoprotein E) that promote or inhibit the fibrosis of α-synuclein, suggesting that altering this balance may be a suitable intervention strategy for PD treatment [24,25,26]. However, studying the mechanism underlying the inhibition of α-synuclein aggregation and preventing the formation of phosphorylated α-synuclein remain hotspots of current research.

Perturbation of metal ion homeostasis is a risk factor for numerous diseases, as the cellular accumulation of metals leads to cytotoxicity, and their depletion markedly affects cell metabolism, as metals act as cofactors for numerous enzymes [27]. Zinc is the second most prevalent trace element after iron and is essential for a wide variety of physiological functions in the human body. In the brain, zinc concentrations are approximately 1.5% of the total content [27]. Increasing evidence suggests that Zn^2+^ accumulation is closely related to the pathogenesis of PD [28] and post-mortem studies have revealed excessive Zn^2+^ deposition in the substantia nigra and striatum of idiopathic patients with PD [29,30,31]. Additionally, in vitro and in vivo experiments using animal models of PD have shown that Zn^2+^ accumulation is a predisposing factor for dopaminergic neuronal loss, α-synuclein aggregation, and impairment of the ubiquitin–proteasome system [32,33,34]. Studies using drug-induced PD animal models such as 1-methyl-4-phenyl-1,2,3,6-tetrahydropyridine or 6-hydroxydopamine have demonstrated that both an increase and decrease in Zn^2+^ concentration damage the dopaminergic neurons, cause oxidative stress, and accelerate apoptosis [35,36]. α-Synuclein presents both high (Asp121) and low (His50) affinity binding sites for Zn^2+^ [37]. However, many studies have confirmed that zinc deficiency also aggravates PD symptoms [27,38,39,40,41]. Clinical meta-analytic studies have also confirmed that the levels of Zn^2+^ in plasma and cerebrospinal fluid of PD patients are lower than those of healthy controls [42,43,44,45,46]. As an important coenzyme for various enzymes and proteins, Zn^2+^ is involved in oxidative stress and inflammatory responses [47,48]. In addition, Zn^2+^ in synaptic vesicles can be released into the synaptic cleft along with glutamate and thus can participate in neurotransmission [49,50]. The degeneration of dopaminergic neurons caused by the accumulation of toxic Zn^2+^ in the cytoplasm is considered a key pathogenic mechanism underlying dopaminergic cell death [35,51]. Excessive accumulation of toxic Zn^2+^ is associated with release from intracellular Zn stores (lysosomes, mitochondria, and metal-binding proteins) and the influx into the extracellular environment [51,52]. Therefore, zinc plays a key role in the pathogenesis of PD and the formation of α-synuclein aggregates.

The maintenance of zinc homeostasis in cells requires the participation of transporters [53]. The PD-related protein, ATP13A2, is reported to utilize ATP to transport inorganic cations, including Zn^2+^ [54]. Additionally, mutations in *ATP13A2*, also known as *PARK9*, which is a gene associated with autosomal recessive forms of early-onset PD, result in the dysregulation of Zn^2+^ homeostasis, thereby promoting lysosomal dysfunction and α-synuclein accumulation [55,56,57,58]. ATP13A2-mediated export of polyamines from late endo/lysosomes into the cytoplasm reduces mitochondrial production of reactive oxygen species (ROS), thereby protecting mitochondrial health and preventing cell death [59,60]. *ATP13A2* encodes a transmembrane lysosomal P-type ATPase that is expressed particularly in nigral dopaminergic neurons and has been identified in Lewy bodies from the brain of sporadic PD patients [61]. It has also been demonstrated to inhibit the aggregation of α-synuclein through lysosome autophagy and/or exosome clearance. It has been reported that ATP13A2 located on multivesicular bodies mainly acts to regulate Zn^2+^ homeostasis in exosomes and influences the exosome-associated externalization of α-synuclein [39]. These studies suggest that an imbalance in Zn^2+^ homeostasis, α-synuclein accumulation, and ATP13A2 (dys)-function are intertwined processes that play a major role in the etiology of PD. However, the underlying mechanisms remain largely unexplored. In this study, we used in vitro and in vivo models of α-synuclein overexpression to analyze Zn^2+^ accumulation, the function of ATP13A2, and related lysosome and mitochondrial factors to unravel the effect of Zn^2+^ on α-synuclein aggregation.

## 2. Results

### 2.1. Administration of Zn^2+^ Reduces Spatial Exploration Behavior and Increases the Expression Levels of ATP13A2 Protein in Brain Tissue of α-Synuclein-GFP Mice

In this study, α-synuclein-GFP transgenic mice were treated with different concentrations of ZnSO_4_ (20 or 200 ppm) through their daily drinking water for three months. To evaluate the effect of ZnSO_4_ treatment, the spatial exploration ability of mice was assessed using an open field test, and the total distance completed during 5 min as well as distance and time of movement in the central area were recorded.

Compared with the control group, mice in the 20 ppm group exhibited a slight reduction in the total distance, but it was not statistically significant; the total distance in the 200 ppm group was significantly shorter than that of the control group (*p* < 0.05) (Figure 1a,d). Additionally, mice in the 20 ppm group presented a shorter movement distance and spent less time in the central zone, although this reduction was not significant. Meanwhile, the mice in the 200 ppm group spent less time and presented a shorter distance in the central area than the control group did (*p* < 0.05) (Figure 1b–d). Furthermore, the anxiety levels (spent less time in the central zone) in the mice of the 200 ppm group were higher than that of mice in the control group, and their ability to explore new environments was relatively weak. Altogether, these results suggested that treatment with 200 ppm ZnSO_4_ significantly reduced the spatial exploration ability and aggravated anxiety in mice.

We then examined the effects of Zn^2+^ treatment on the ion levels in the brains of α-synuclein-GFP transgenic mice. Using ICP-MS, we confirmed that administration of ZnSO_4_ through the drinking water increased the level of Zn^2+^ in the brain of α-synuclein-GFP transgenic mice. In the 20 ppm group, the concentration of Zn^2+^ was not significantly different than that of the control group. However, the concentration of Zn^2+^ in the brains of the 200 ppm group was significantly higher than that of the control (Figure 1e).

To investigate whether Zn^2+^ affected α-synuclein aggregation, we next examined the effects of ZnSO_4_ on the expression levels of ATP13A2. ATP13A2 expression levels in the 200 ppm group were significantly higher than those in the control group (Figure 1f,g). Immunohistochemical staining revealed that ATP13A2 was mainly located in the cytoplasm of the cells in the substantia nigra pars compacta. The positive staining intensity of ATP13A2 in the 200 ppm group was higher and stronger than that in the control group (Figure 1h). Altogether, these results indicated that high concentrations of Zn^2+^ affected the expression of ATP13A2 in the brain tissue of α-synuclein-GFP transgenic mice.

### 2.2. Zn^2+^ Aggravates the Pathology of α-Synuclein-GFP Transgenic Mice by Increasing the Expression Levels of α-Synuclein and Decreasing the Expression Levels of Tyrosine Hydroxylase (TH)

Tyrosine hydroxylase (TH) catalyzes the rate-limiting step in the synthesis of dopamine [62]. Hence, the lack of TH notably affects dopamine synthesis. Based on this, we aimed to evaluate whether Zn^2+^ imbalance altered the expression levels of TH. Immunohistochemistry staining revealed that TH levels were significantly lower in the substantia nigra pars compacta of mice in the 200 ppm group than those in the control mice. However, TH levels in mice in the 20 ppm Zn^2+^ did not significantly differ from those of the control group (Figure 2a,b). Western blot analysis also revealed that 200 ppm of Zn^2+^ significantly reduced TH expression compared to that in the control group (Figure 2c,d).

Then, we aimed to evaluate the aggregation of α-synuclein by examining the green fluorescence in α-synuclein-GFP transgenic mice. The α-synuclein-GFP transgenic mice exhibited green fluorescence, confirming that the model had been established correctly. Additionally, the green fluorescence was increased in the 200 ppm group compared with the control group (Figure 2e). According to Western blot results, the expression levels of endogenous α-synuclein and phosphorylated (p)-α-synuclein in the substantia nigra of mice in the 200 ppm group were significantly higher than those in the control group, suggesting that this may be one of the mechanisms through which zinc triggers neurotoxicity and contributes to the pathogenesis of PD. However, the expression levels of the exogenous human α-synuclein-GFP did not change significantly in any of the groups. In addition, the expression levels of endogenous α-synuclein, human α-synuclein-GFP, and p-α-synuclein in the 20 ppm group showed no obvious changes (Figure 2f–j). Taken together, the results indicated that treatment with 200 ppm of Zn^2+^ damaged dopaminergic neurons in the substantia nigra of mice.

### 2.3. Zn^2+^ Treatment Downregulates the Autophagy–lysosome Pathway and Accelerates the Apoptosis Pathway in α-Synuclein-GFP Transgenic Mice

The degradation of α-synuclein in cells occurs mainly through the ubiquitin-mediated proteasome pathway and the autophagy–lysosomal pathway [63]. Hence, we examined the changes in the expression levels of autophagy- and ubiquitin-related proteins. The expression levels of lysosomal proteins LAMP-1, LAMP-2a, and the ratio of LC3II to LC3I in the substantia nigra of mice in the 200 ppm group were significantly lower than those in the control group. However, the 20 ppm group only exhibited a decrease in the levels of LAMP-2a and the ratio of LC3II to LC3I, but no change was observed in LAMP-1 expression (Figure 3a–c,e). Moreover, compared with the control group, there was no significant difference in the expression levels of the P62 protein, which is a classical receptor of autophagy and a ubiquitin sensor, in the 20 ppm group compared with the control group, while the expression levels of P62 protein in the 200 ppm group were significantly increased (Figure 3d).

Caspase 3 exists in cells in the form of zymogen, which is cleaved to activate apoptosis signaling and initiate Caspase-dependent apoptosis. Apoptotic signaling also causes the release of cytochrome C from mitochondria into the cytoplasm [64]. In addition, apoptosis is modulated by multiple phosphorylations of several different protein kinases, including the major subtypes of mitogen-activated protein (MAP) kinase, i.e., ERK1, ERK2, and p38 kinase [65].

As shown in Figure 3, the cytoplasmic cytochrome C protein levels (Figure 3f,j) and the phosphorylation levels of ERK1 (Figure 3f,g), ERK2 (Figure 3f,h), and p38 (Figure 3f,i) in the substantia nigra of the 20 and 200 ppm mice were significantly increased compared to those in the control group. Moreover, the protein levels of activated caspase 3 (Figure 3f,k) were significantly increased in the 200 ppm group, but not in the 20 ppm group compared to those in the control group. Altogether, these results suggested that treatment with 200 ppm of Zn^2+^ caused caspase-dependent apoptosis in mouse substantia nigra cells, while treatment with 20 ppm of Zn^2+^ only increased the cytochrome C levels in the cytoplasm. Both 20 and 200 ppm Zn^2+^ treatments increased the phosphorylation levels of ERK, which activates the P38 signaling pathway and induces apoptosis.

### 2.4. Zn^2+^ Upregulated Atp13A2 and α-Synuclein Protein Levels in HEK293 α-Synuclein-DsRed Cells, While It Inhibited the Autophagy–lysosome Pathway

The effect of Zn^2+^ on the viability of HEK293 α-synuclein-DsRed cells was assessed using the MTS assay. For this purpose, HEK293 α-synuclein-DsRed cells were treated with 10 and 75 μM ZnSO_4_ for 24 h according to the results of the MTS assay (Figure 4a). Staining with N-(6-methoxy-8-quinolyl)-p-toluenesulfonamide (TSQ), which specifically stains Zn^2+^, showed that the fluorescence intensity was significantly increased in both treatment groups compared with the control group and that the fluorescence of the cells treated with 75 μM ZnSO_4_ was stronger than that of cells treated with 10 μM ZnSO_4_ (Figure 4b). This indicated that the established approach was valid, as the intracellular Zn^2+^ content increased after ZnSO_4_ administration. Moreover, double labeling immunofluorescence and subsequent confocal microscopy results showed that α-synuclein levels (red fluorescence) and ATP13A2 levels (green fluorescence) were significantly increased in cells treated with 75 μM ZnSO_4_ compared with the control group (Figure 4c). In addition, the expression levels of ATP13A2, α-synuclein, and p-α-synuclein were significantly increased in cells treated with 75 μM ZnSO_4_ compared with the control groups. Moreover, the expression levels of α-synuclein were also significantly increased in the 10 μM ZnSO_4_ group compared with the control group (Figure 4d–h).

Next, we aimed to assess whether the observed increase in α-synuclein protein levels resulted from a decrease in the expression of autophagy-related proteins. As shown in Figure 4, the expression levels of LAMP-2a and the ratio of LC3II to LC3I were decreased and the expression levels of P62 were increased after treatment with 75 μM ZnSO_4_ compared with the control group (Figure 4i,k–m). The expression levels of LAMP1 did not change significantly in any of the groups (Figure 4i,j). The expression levels of these proteins did not change significantly in the group treated with 10 μM ZnSO_4_ compared with the control group (Figure 4i–m). These results suggested that treatment with 75 μM ZnSO_4_ caused abnormal lysosomal autophagy in HEK293 α-synuclein-DsRed cells.

### 2.5. Zn^2+^ Activates the ERK/P38 Signaling Pathway and Causes Mitochondrial Damage in HEK293 α-Synuclein-Dsred Cells

Cadmium can induce apoptosis by activating the ERK/MAPK and P38/MAPK pathways, which ultimately activate mitochondria-mediated intrinsic apoptosis [66]. We hypothesized that zinc may also exert the same effect. To address this, we examined the phosphorylation levels of ERK1, ERK2, and P38 proteins in cells treated with 10 and 75 μM ZnSO_4_ for 12 h. Compared with the control group, the phosphorylation levels of ERK1, ERK2, P38, and the protein expression levels of cytochrome C and active caspase 3 were increased in the 75 μM ZnSO_4_ group. However, after the cells were treated with 10 μM ZnSO_4_, the expression levels of proteins mentioned above were not significantly different from those in the control group (Figure 5a–f). These results were consistent with the results of animal experiments and supported the hypothesis that ZnSO_4_ induced apoptosis through the ERK/P38 pathway.

Next, we examined whether apoptosis induced by the ERK/P38 signaling pathway leads to impairment of mitochondrial function. Mitochondrial complexes I and III are important components of the mitochondrial respiratory chain. Deletion of the mitochondrial complex I and III subunits affects the enzymatic activity of the complexes, which in turn blocks electron transport through the respiratory chain and leads to the accumulation of ROS [67,68]. We selected a total of 12 important subunits from complexes I and III [69,70] and assessed their expression levels using RT-qPCR after treating the cells with 75 μM ZnSO_4_. Treatment with 75 μM ZnSO_4_ remarkably decreased the mRNA expression levels of five subunits of mitochondrial complex I, namely NDUFA2, NDUFA9, NDUFB7, NDUFB10, and NDUFS5; however, the mRNA expression levels of the other subunits analyzed remained unaltered (Figure 5g).

Finally, to further explore the effect of Zn^2+^ on mitochondrial function, we evaluated mitochondrial membrane potential in cells using JC-1. JC-1 is a fluorescent substance that can selectively enter cells and bind to mitochondria, which can be used to detect mitochondrial membrane potential [71]. The results (Figure 5h) showed that compared with the control group, the red fluorescence was reduced and the green fluorescence was enhanced in the 75 μM Zn^2+^ group, indicating a decrease in mitochondrial membrane potential.

Altogether, these results indicated that treatment with 75 μM Zn^2+^ reduced the mRNA expression levels of some of the subunits of mitochondrial complex I and decreased mitochondrial membrane potential, thereby attenuating the release of mitochondrial respiratory chain enzymes and impairing mitochondrial function.

### 2.6. Overexpression of ATP13A2 Inhibits Zinc-Induced Autophagy–Lysosome Pathway Inhibition and ERK/P38 Signaling Pathway Activation

ATP13A2 is involved in the regulation of Zn^2+^ homeostasis in cells, and loss of ATP13A2 causes an imbalance in Zn^2+^ homeostasis, thereby promoting abnormal aggregation of α-synuclein [58]. To further verify the role of ATP13A2, we overexpressed it in HEK293 α-synuclein-DsRed cells and then treated with 75 μM Zn^2+^. First, we verified the successful construction of the HEK293 α-synuclein-DsRed cell line overexpressing ATP13A2. As shown in Figure 6, the expression levels of ATP13A2 protein in HEK293 α-synuclein-DsRed cells transfected with ATP13A2 overexpression plasmid were significantly higher than those in cells transfected with the empty vector (Figure 6a,b). Treatment with 75 μM ZnSO_4_ in cells overexpressing ATP13A2 inhibited the significant increase in p-α-synuclein, α-synuclein, and P62 expression levels caused by Zn^2+^ (Figure 6c–e,g) and alleviated the significant decrease in expression levels of LAMP-2a and the ratio of LC3II to LC3I (Figure 6c,f,h) that were observed when cells transfected with the empty vector were treated with ZnSO_4_.

Next, we examined the expression changes of proteins in the ERK/P38 signaling pathway. Our results showed that the expression levels of p-ERK1, p-ERK2, p-P38, cytochrome C, and cleaved caspase 3 in the cells of the ATP13A2-overexpressing group were not significantly changed upon treatment with 75 μM ZnSO_4_ (Figure 6i–n). Altogether, these results suggest that ATP13A2 could inhibit the inhibition of the autophagy–lysosome pathway and activation of the ERK/P38 signaling pathway induced by Zn^2+^.

### 2.7. ATP13A2 siRNA Promotes Zinc-Induced Activation of ERK/P38 Signaling Pathway

We further explored the role of ATP13A2 in the aggregation of α-synuclein induced by Zn^2+^ treatment by transfecting HEK293 α-synuclein-DsRed cells with siRNA targeting ATP13A2 for 72 h. Western blot and RT-qPCR results confirmed that ATP13A2 silencing was effective, as 72 h post-transfection, the mRNA levels of ATP13A2 decreased by 69% compared to the cells in the negative control group. Additionally, the ATP13A2 protein levels decreased by 45% in the cells treated with siRNA compared with the negative control group (Figure 7a–c).

In the absence of ZnSO_4_ treatment, the p-α-synuclein protein expression levels did not change significantly after silencing ATP13A2 compared with the negative control group. Treatment with 75 μM ZnSO_4_ resulted in a significant increase in the expression levels of p-α-synuclein compared with the control group; this increase was more obvious in the silenced ATP13A2 group (Figure 7d,e).

Finally, we examined the changes in the expression levels of proteins in the ERK/P38 signaling pathway. In both the ATP13A2 siRNA group and the negative control group, upon treatment with 75 μM ZnSO_4_, the phosphorylation levels of ERK1, ERK2, P38, and cleaved caspase 3 were significantly increased, while expression levels of LAMP-2a were decreased compared with the control group. Moreover, the phosphorylation levels of ERK2 and P38 were more significantly elevated in cells with silenced ATP13A2 (Figure 7f–k). Altogether, these results indicated that ATP13A2 silencing did not lead to increased expression of p-α-synuclein and apoptosis in HEK293 α-synuclein-DsRed cells without the involvement of Zn^2+^; however, in the presence of Zn^2+^, ATP13A2 silencing increased p-α-synuclein and activated the ERK/P38 signaling pathway to induce apoptosis. This further suggested that ATP13A2 inhibited Zn^2+^-induced aggregation of α-synuclein with a protective effect.

## 3. Discussion

In previous studies, excessive Zn^2+^ was reported to cause reduced TH-positive neurons and promote the aggregation of α-synuclein in PD models [29,34]. In the present study, we found that the expression of ATP13A2 increased when the intracellular Zn^2+^ concentrations were high. As a P5-type ATPase expressed in acidic vesicles, ATP13A2 plays an important role in regulating Zn^2+^ and is especially sensitive to mitochondrial apoptosis and lysosomal autophagy in the process of clearing the α-synuclein accumulation induced by Zn^2+^ [57,60,72].

In this study, we used a mouse transgenic line stably expressing human α-synuclein fused to GFP, which has been widely used to study how PD risk factors or therapeutic compounds affect α-synuclein aggregation and motor impairment caused by α-synuclein dysregulation [73]. It was observed that α-synuclein-GFP accumulation and aggregation correlated with progressive behavioral deficits in α-synuclein-GFP mice. Additionally, we found that ATP13A2 plays an important regulatory role in zinc-induced cellular damage and α-synuclein phosphorylation levels [58,74,75,76]. Wild-type ATP13A2 is localized on the lysosome and is involved in clearance and ion transport, whereas the mutant forms that are associated with PD are localized to the endoplasmic reticulum, resulting in the aberrant accumulation of α-synuclein [18,54,77,78]. In previous studies using clinical samples, it was found that ATP13A2 levels are substantially lower in post-mortem tissue biopsies of patients with sporadic PD but are higher in the cytoplasm of surviving dopamine neurons, where the protein accumulates in Lewy bodies [61,79]. ATP13A2 present in multivesicular bodies can participate in the transport of zinc in multivesicular bodies [54], while increased ATP13A2 expression reduces intracellular α-synuclein levels via increased externalization of exosome-associated α-synuclein [39]. This was consistent with our results, showing that increased α-synuclein levels induced by high concentrations of Zn^2+^ stimulated the expression of ATP13A2. Our results confirmed that ATP13A2 plays an important regulatory role in the maintenance of Zn^2+^ homeostasis in PD.

Recently, it has been reported that the loss of ATP13A2 leads to altered homeostasis of Zn^2+^, which might lead to the blockage of Zn^2+^ transport in lysosomes and affect the abnormal function of proteins related to lysosome function [58,80,81]. In our experimental results, exogenous Zn^2+^ stimulation increased the expression of ATP13A2. The main reason for this could be the role of ATP13A2 in the forward transport of Zn2+ and the increased clearance of abnormally phosphorylated α-synuclein. Double-(ATP13A2^−/−^ and α-synuclein overexpressing) mutant mice presented histopathological characteristics of similar severity as mice harboring only the ATP13A2^−/−^ mutation [82]. Heterozygous GBA D409V and ATP13A2 mutations do not exacerbate pathological α-synuclein development in the prodromal preformed fibril model in young mice [83]. These results were consistent with our findings, indicating that deletion of ATP13A2 alone did not result in increased α-synuclein protein levels. ATP13A2 affected the phosphorylation levels of α-synuclein only in the presence of Zn^2+^. After increasing ATP13A2 expression using transient transfection, in vitro experiments confirmed that ATP13A2 inhibited the phosphorylation of α-synuclein, enhanced the expression levels of LAMP2, and decreased the expression levels of apoptosis-related factors. To a certain extent, these results explained the effect of ATP13A2 on the maintenance of zinc homeostasis, indicating that it could regulate Zn^2+^ concentrations to reduce cytotoxicity.

In our study, we found that abnormal aggregation of α-synuclein caused by excessive Zn^2+^ is related to LAMP2, which is a major component of chaperone-mediated autophagy and is directly associated with the clearance of α-synuclein [84,85]. From our results, it can be concluded that Zn mainly affects the expression of LAMP-2 in lysosomes, which affects the protein expression of downstream P62 and LC3. Under high intracellular Zn^2+^ conditions, the overexpression of ATP13A2 reversed the reduction in the expression levels of LAMP-2, indicating that ATP13A2 protected the function of lysosomes by regulating Zn^2+^ concentrations. Previous studies have confirmed that ATP13A2 is not only present in lysosomes but also acts as a zinc transporter located on the membranes of multivesicular bodies [54,57,86]. ATP13A2 plays an important role in the maintenance of endogenous zinc concentration in vesicles, as well as in the clearance of α-synuclein aggregates through encapsulation into exosomes and its subsequent excretion [60]. Elevated ATP13A2 expression reduces intracellular α-synuclein levels via increased externalization of exosome-associated α-synuclein, which could explain why the surviving neurons of the substantia nigra pars compacta in sporadic PD patients exhibit ATP13A2 overexpression [39]. Although we did not observe the localization of ATP13A2 in multivesicular bodies under high Zn conditions, the endogenous ATP13A2 helped cells maintain Zn transport and regulate α-synuclein by altering the expression of LC3 and P62 and by weakening the damage to the mitochondrial oxidative respiratory chain. The exogenous ATP13A2 prevented the dysfunction of mitochondrial lysosomes caused by Zn^2+^.

Due to the toxic effect of excessive Zn^2+^ on lysosomes and mitochondria, the mitochondrial oxidative respiratory chain was also damaged, resulting in α-synuclein accumulation. The endogenous apoptotic pathway triggered by mitochondrial damage is an important mechanism of apoptosis in PD. In this experiment, 200 ppm ZnSO_4_ in vivo and 75 μM ZnSO_4_ in vitro caused an increase in cytochrome C protein expression, activation of the ERK/P38 signaling pathway, and activation of caspase 3. Zn^2+^ can act as a pro-apoptotic signal, can activate the P38 signaling pathway, and can promote the release of cytochrome C on the mitochondrial membrane into the cytoplasm as an intracellular zymogen to activate caspase 3, thereby causing apoptosis [87]. Recent studies have shown that the c-isoform of mitochondrial ATP synthase is abnormally aggregated in lysosomes in mice with brain-specific deletion of *ATP13A2* and is involved in the pathogenesis of lipofuscinosis in PD [88]. The promoter region of human *ATP13A2* contains a hypoxia response element that binds to hypoxia-inducible factor 1α (HIF-1α), leading to the transcription of the gene [89,90]. *ATP13A2* expression increased in the HEK293 cells and dopaminergic MN9D cells under hypoxic conditions, indicating that ATP13A2 can resist oxidative stress and protect mitochondrial function [90]. *ATP13A2* knockout resulted in abnormal mitochondrial morphology and induced an increase in ROS in SH-SY5Y cells [60,91,92]. In the present study, HEK293 cells overexpressing ATP13A2 were treated with 75 μM Zn^2+^, and the results showed that the cytochrome C levels did not change significantly, and the ERK/P38 signaling pathway and caspase 3 were not activated. These results indicate that ATP13A2 protects cells from apoptosis caused by zinc imbalance. Many experiments on the regulation of zinc homeostasis by ATP13A2 are carried out by changing the expression of ATP13A2 to study the underlying mechanism. Our goal was to establish a high zinc environment, observe the changes in ATP13A2 protein and α-synuclein expression levels in cells and animals, and demonstrate the coordination between endogenous proteins. Our results confirmed that zinc cytotoxicity directly affects lysosomal and mitochondrial functions, resulting in an increase in phosphorylation of α-synuclein, wherein endogenous ATP13A2 plays a key regulatory role. However, further studies are still necessary to explore the relationship between the regulation of ATP13A2 and other zinc transporters and to investigate whether ATP13A2 has an inhibitory effect on the production of α-synuclein in the presence of zinc.

In conclusion, the results of the present study demonstrated that high Zn concentrations are cytotoxic and induce both autophagy and apoptosis in α-synuclein-induced PD models. ATP13A2 modulated lysosomal and mitochondrial functions by suppressing zinc cytotoxicity. ATP13A2 plays an important role in the phosphorylation of pathologic α-synuclein in a Zn^2+^-induced α-synucleinopathy model. These results suggest the possibility that ATP13A2 has protective effects against the abnormal aggregation of α-synuclein, lysosomal dysfunction, and apoptosis caused by Zn^2+^ and that zinc homeostasis may be a useful treatment strategy for neurodegenerative disorders. Finally, our results contribute to the existing body of knowledge regarding the relationship between ATP13A2 and zinc homeostasis.

## 4. Materials and Methods

### 4.1. Animals and Treatment

The nine-month-old α-synuclein-GFP mouse models (courtesy of Professor Jiayi Li) were selected and randomly divided into control (Ctrl), ZnSO_4_ 20 ppm (20 ppm), and ZnSO_4_ 200 ppm groups (200 ppm), with 10 mice in each group, reared in the same environment. They were housed under a light/dark cycle of 8:00/20:00 and controlled temperature (24 ± 2 °C) and humidity (40–70%) conditions. ZnSO_4_ was added to the drinking water of the mice in the zinc group to the final concentrations of 20 and 200 ppm. The mice in the control group were given normal drinking water. All mice were fed with drinking water for three months and sacrificed after behavioral testing. All treatments and experimental procedures were performed in accordance with the National Institutes of Health guidelines and approved by the Northeastern University Laboratory Animal Ethical Committee. Table 1 summarizes the reagent information.

### 4.2. Cell Cultures and Treatments

Human embryonic kidney epithelial cells (HEK-293 T) stably overexpressing α-synuclein-DsRed (HEK293 α-synuclein-DsRed) were donated by Professor Jiayi Li (Wallenberg Neuroscience Center, Lund, Sweden). Cells were grown (at 37 °C and 5% CO_2_) on 6 cm tissue culture dishes (1 × 10^6^ cells per dish) in appropriate media. In each experiment, cells were grown in a serum-free medium for an additional 24 h prior to zinc treatment, and then incubated with 10 or 75 μM ZnSO_4_ for 12 h. We cloned the ATP13A2-HA plasmid; the *ATP13A2* sequence (NM_022089.4) was obtained from GenBank. After *ATP13A2* was double digested with restriction enzymes Hind III and Xho I, it was inserted into pcDNA3.1-HA vector for sequencing. *ATP13A2*-pcDNA3.1-HA and pcDNA3.1-HA were transiently transfected into the HEK293 α-synuclein-DsRed cell line using the recommended amount of lipofectamine 2000. The siRNA of *hATP13A2* (NM_022089.4) was synthesized and prepared by Shanghai Gema Pharmaceutical Technology Co., Ltd. The sequences of the for *hATP13A2* mRNA are 5′-GGAUCCCUUUGCUGCUCUUTT-3′ and 5′-AAGAGCAGCAAAGGGAUCCTT-3′. All transfection experiments were performed using the recommended volumes of lipofectamine 2000. The effects of *ATP13A2* overexpression and gene silencing in HEK293 α-synuclein-DsRed cells were demonstrated using real-time PCR and Western blot analysis.

### 4.3. Open Field Test

The open field test (OFT) is a typical method used to observe spontaneous activity, exploratory behaviors, and tension in experimental animals in a novel environment. All mice in the three groups were placed in the center of the open field for 30 min to launch the test. The mice were placed in the center of the field in a quiet environment, and the digital camera continuously recorded their behavior for 5 min. According to the experimental requirements, the experimental parameters such as the total movement distance, the central movement distance, and the time spent in the center zone were analyzed and collected using the SMART 3.0 software. To prevent interference between trials, the inner wall and bottom of the arena were cleaned during the inter-trial intervals [93].

### 4.4. Western Blot Analysis

Tissues and cells were lysed in RIPA buffer (50 mmol/L Tris-HCl (pH 7.5), 150 mmol/L NaCl, 1% NP-40, 1% sodium deoxycholate, 0.1% SDS, 25 mmol/L NaF, 5 mmol/L EDTA, 2 mmol/L Na_3_VO_4_, and 1 mmol/L PMSF) containing 1:100 diluted protease inhibitor cocktail (Sigma, Burlington, MA, USA) for 30 min on ice. For extraction of all proteins in mice SN tissues or cells, the lysates were centrifuged at 12,000× *g* for 30 min at 4 °C. The protein concentrations in the lysates were determined using a BCA protein assay kit (Thermo Scientific-Pierce, Waltham, MA, USA). The supernatants containing 10 μg protein were subjected to 4–12% SDS-PAGE for separation and then transferred onto polyvinylidene fluoride (PVDF) membranes (Millipore, Burlington, MA, USA). The membranes were incubated with 5% BSA solution at room temperature for 1 h. Subsequently, the membranes were respectively incubated with primary antibodies overnight at 4 °C. The primary antibodies required for this experiment are shown in Table 2. Finally, the membranes were incubated with the horseradish peroxidase (HRP)-conjugated secondary antibodies for 2 h after washing. Bands were detected using a chemiluminescence imaging analysis system (Tanon, 5500, Shanghai, China) and enhanced chemiluminescence (ECL) Kits (EMD Millipore, Burlington, MA, USA). Each experiment was repeated at least 3 times.

### 4.5. Immunohistochemistry

The mice in the three experimental groups (*n* = 6 in each group) were anesthetized and half brains were removed and fixed using 4% paraformaldehyde and then embedded in paraffin. Briefly, serial 5 µm coronal sections were preincubated with 5% bovine serum albumin [BSA] and 1% normal goat serum for 1 h and then incubated overnight at 4 °C in a humidified chamber with rabbit monoclonal anti-ATP13A2 and anti-TH antibodies (1:200, Table 2). Next, the sections were treated with biotinylated goat anti-mouse IgG (1:500) for 1 h at room temperature and then processed with the avidin–biotin–peroxidase (ABC) complex (1:100). Finally, the sections were developed in 3,3′-diaminobenzidine. Incubation of one section in normal mouse serum (1:100) served as a negative control for nonspecific staining. Images of immunohistochemical staining were captured using a light microscope (DM4000B; Leica, Wetzlar, Germany).

### 4.6. Immunofluorescence Staining and Confocal Laser Scanning Microscopy

Immunofluorescence staining was performed as described in our previous experiment with certain modifications. HEK293 α-synuclein-DsRed cells were cultivated on coverslips coated with poly-D-lysine (Sigma-Aldrich, Burlington, MA, USA) in 6-well plates. After treatment with 75 μM ZnSO_4_ for 12 h, cells were washed with PBS, fixed with ice-cold methanol for 30 min, and then rehydrated with PBS and treated with 5% goat serum/PBS for 1 h. The sections were incubated with a mixture of rabbit anti-ATP13A2 antibody (1:500, abcam, Cambridge, UK) and anti-α-synuclein monoclonal antibody (1:500, Novus) overnight at 4 °C. The sections were treated with Alexa Fluor^®^ 488-and Alex Fluor^®^ 594-conjugated secondary antibodies fluorescence after rinsing and finally labeled using DAPI. The sections were mounted with an antifade mounting medium. Images were taken using the laser scanning confocal microscope (Leica, TCS, SP8, Wetzlar, Germany). The details regarding the antibodies used for immunofluorescence staining are presented in Table 2.

### 4.7. Zinc Analysis with Inductively Coupled Plasma Mass Spectrometry (ICP-MS)

First, 100 mg brains of mice in the three experimental groups were used to measure zinc levels, and samples were digested in 90% HNO_3_ at 105 °C for 30 min. Then, samples were diluted to the appropriate multiple and measured using 7500a-ICP-MS (Aglient Technologies Inc., Santa Clara, CA, USA). Isotopes Zn^66^ were chosen for optimal sensitivity of quantification in data acquisition mode (spectral analysis).

### 4.8. Zinc Staining

The procedure and method used for *N*-(6-methoxy-8-quinolyl)-p-toluenesulfonamide (TSQ) staining was as described previously. Briefly, HEK293 α-synuclein-DsRed cells were cultured in 12-well plates, treated with zinc, and immersed in 4.5 μM TSQ solution for 1 min. Fluorescence microscopy (Nikon, Ds-Ri1, Melville, NY, USA) was used to observe the zinc reaction.

### 4.9. Quantitative Real-Time Polymerase Chain Reaction (qPCR)

Total RNA was extracted from HEK293 α-synuclein-DsRed cells using TRIzol reagent (Invitrogen) according to the manufacturer’s instructions, and 500 ng template RNA was reverse transcribed into cDNA using the GoScript™ Reverse Transcription System (Promega, A5001). Using the Bio-Rad CFX PCR system, PCR reactions were performed in triplicate at a volume of 10 μL reaction mixture and using 20 ng cDNA templates. The sequences of the genes encoding GAPDH and mitochondrial complexes I and III were obtained from the GenBank database, and specific primers were designed using Primer Premier 5.0 (Table 3). The mRNA expression was calculated using the 2^−ΔΔCT^ method.

### 4.10. JC-1 Staining

Here, 75 μM ZnSO_4_-treated or untreated HEK293 α-synuclein-DsRed cells were stained with JC-1 (Invitrogen, Carlsbad, CA, USA) according to the manufacturer’s instructions. Briefly, cells were incubated with 5 μM of JC-1 for 20 min (37 °C, 5% CO_2_), and then, cells were washed and observed under the fluorescence microscope. JC-1 forms J-aggregates to emit orange/red fluorescence at normal or high mitochondrial membrane potential, while JC-1 exhibits green fluorescence in monomeric form when the membrane potential is depolarized.

### 4.11. Cell Viability Assay

CellTiter 96^®^ AQueous One Solution Cell Proliferation Assay (MTS, Promega, Madison, WI, USA) was conducted to evaluate the effect of Zinc on the proliferation and viability of the HEK293 α-synuclein-DsRed cells. First, 1 × 10^4^ HEK293 α-synuclein-DsRed cells per well were seeded in 96-well plates and treated with different concentrations of zinc (10, 50, 75, 100, 150 μM) for 12 and 24 h at 37 °C. Subsequently, 20 μL of MTS was added to each well and further incubated for 4 h. The absorbance was measured at 490 nm in a microplate reader (Bio-Rad, Laboratories, Inc., Hercules, CA, USA).

### 4.12. Statistical Analysis

Data are presented as mean ± SEM. Student’s *t* test or one-way ANOVA were used to analyze differences between groups, as appropriate. Two-way ANOVA analysis was used to analyze the multiple groups and treatments. The analyses were performed using ImageJ and GraphPad Prism 8.0 software. *p* < 0.05 or *p* < 0.01 were considered statistically significant.

## Figures and Tables

**Figure 1 ijms-23-08035-f001:**
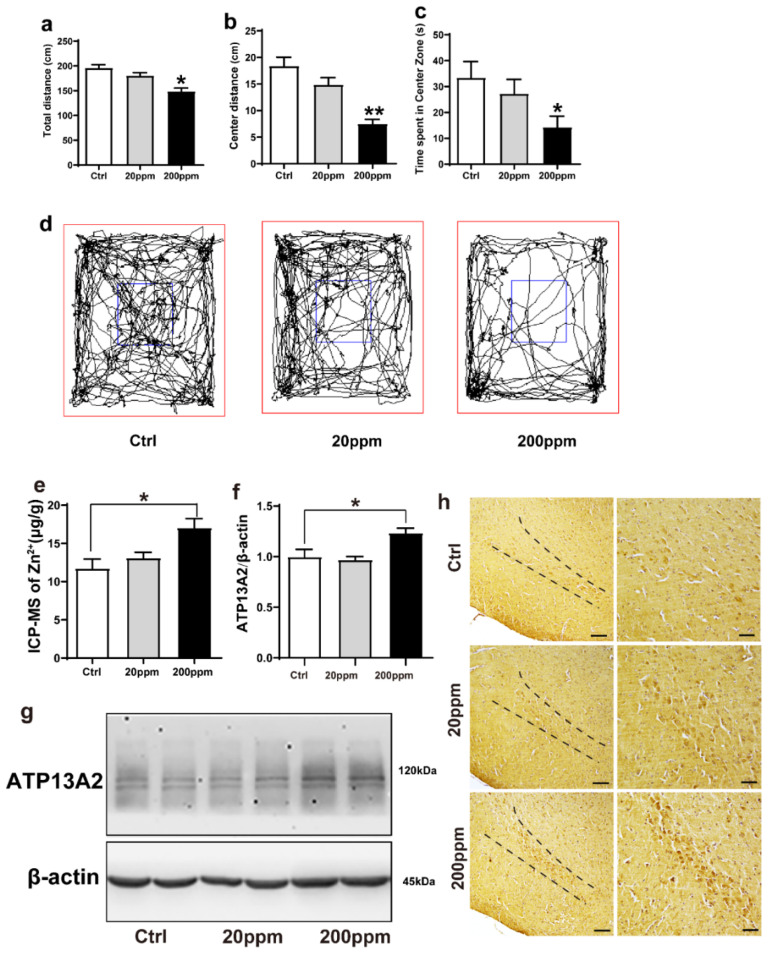
Zn^2^^+^ treatment reduces spatial exploration behavior and increases the expression levels of ATP13A2 protein in the brains of mice. The nine-month-old α-synuclein-GFP mouse models were treated with different doses of ZnSO_4_ (20 or 200 ppm) for three months. (**a**) Total distance traveled in open field exploration in 5 min. (**b**) Movement distance in zone center showing 5 min of open field exploration by the mice. (**c**) Time spent in the center zone showing 5 min of open field exploration by the mice. (**d**) Five minutes movement tracks of open field exploration by the mice. ((**a**–**d**), *n* = 10 mice per group) (**e**) Zinc levels in the brain tissue were determined using ICP-MS ((**e**), *n* = 6 mice per group), and expression of the ATP13A2 (**g**), representative pictures; (**f**), quantification; *n* = 6 mice per group) was analyzed using immunoblotting. (**h**) ATP13A2 positive reaction was detected in substantia nigra pars compacta cells via immunohistochemistry using ATP13A2 antibody (scale bar = 50 μm) ((**h**), *n* = 6 mice per group). Data are represented as the means ± SEM. The maximum and minimum values were removed from each group. Compared with the control group * *p* < 0.05, ** *p* < 0.01.

**Figure 2 ijms-23-08035-f002:**
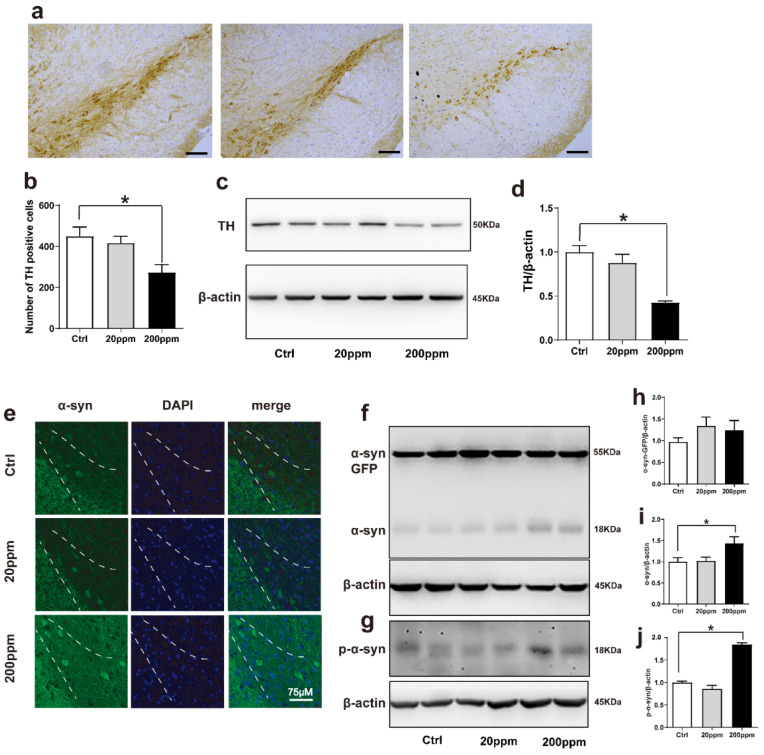
Zn^2+^ aggravates the pathology of α-synuclein-GFP transgenic mice. Zn^2+^ aggravated the pathology of α-synuclein-GFP transgenic mice by increasing the expression of α-synuclein and decreasing the expression of TH. (**a**,**b**) Paraffin sections were stained with TH in the SN to investigate dopamine synthesis and to perform a quantitative analysis (Scale bars = 50 μm). (**c**–**j**) TH, α-synuclein-GFP, α-synuclein and p-α-synuclein levels were detected by Western blotting and quantitative analysis (**c**,**f**,**g**) representative pictures; (**d**,**h**–**j**) quantification). (**e**) The distribution of green-labeled α-synuclein in the substantia nigra was observed using immunofluorescence (Scale bar = 75 μm). Data are represented as the means ± SEM (*n* = 6). Compared with the control group * *p* < 0.05.

**Figure 3 ijms-23-08035-f003:**
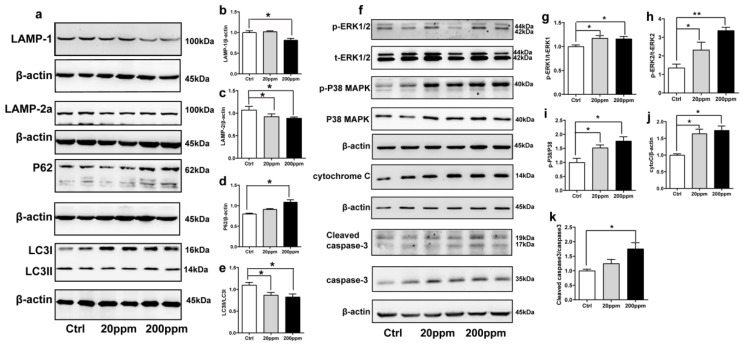
Effects of Zn^2+^ on autophagy–lysosome and apoptosis pathways in mice. Autophagy–lysosome pathway ((**a**), representative pictures; (**b**–**e**), quantification): the expression levels of LAMP-1 (**a**,**b**), LAMP-2a (**a**,**c**), P62 (**a**,**d**), LC3 I and LC3 II ((**a**,**e**) quantification of ratio of LC3II to LC3I) were determined through Western blot analysis using β-actin as an internal control. Apoptosis pathway ((**f**), representative pictures; (**g**–**k**), quantification): the p-ERK1 (**f**,**g**), p-ERK2 (**f**,**h**), p-p38 (**f**,**i**), cleaved caspase 3 (**f**,**k**) levels and the total protein levels of ERK1/2, p38, caspase 3 and expression levels of cytochrome C (**f**,**j**) were determined using Western blotting with β-actin as an internal control. Data are represented as the means ± SEM of at least three independent experiments (*n* = 6). * *p* < 0.05, ** *p* < 0.01 with respect to the control group.

**Figure 4 ijms-23-08035-f004:**
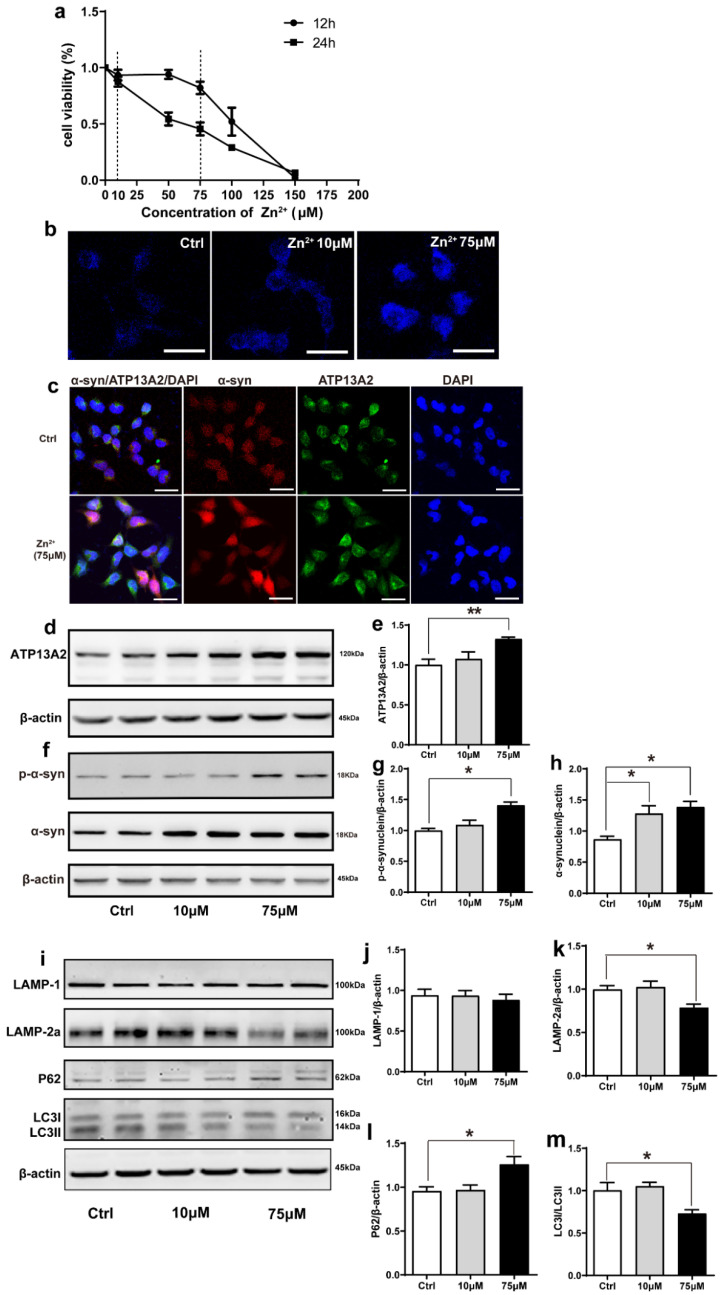
Effects of Zn^2+^ on ATP13A2, α-synuclein, and autophagy–lysosome pathway in HEK293 α-synuclein-DsRed cells. (**a**) Cell viability of HEK293 α-synuclein-DsRed cells incubated with different concentrations of Zn^2+^ for 12 and 24 h. All values are presented as the mean ± standard error of the mean (SEM) of at least three independent experiments. (**b**) TSQ staining was used to verify the zinc fluorescence intensity in HEK293 α-synuclein-DsRed cells with a medium containing 10 or 75 μM ZnSO4. Bar = 25 μm. (**c**) HEK293 α-synuclein-DsRed cells were untreated or treated with 75 μM ZnSO4 for 12 h. Cells were observed using confocal microscopy after staining with ATP13A2 and DAPI. The spontaneous red fluorescence of HEK293 α-synuclein-DsRed cells represented the expression levels of α-synuclein. Scale bar = 25 μm. (**d**–**m**) HEK293 α-synuclein-DsRed cells were treated with 10 or 75 μM ZnSO4 for 12 h. Immunoblot images ((**d**,**f**), left panels) and quantification ((**e**,**g**,**h**), right panels) show the expression of ATP13A2 (**d**), p-α-synuclein and α-synuclein (**f**) in the HEK293 α-synuclein-DsRed cells. Autophagy–lysosome pathway ((**i**), representative pictures; (**j**–**m**) quantification): the expression levels of LAMP-1 (**i**,**j**), LAMP-2a (**i**,**k**), P62 (**i**,**l**), LC3 I and LC3 II ((**i**,**m**), quantification of ratio of LC3II to LC3I) were determined using Western blot analysis with β-actin as an internal control. All data from at least three independent experiments are represented as the means ± SEM. * *p* < 0.05, ** *p* < 0.01 with respect to the control group.

**Figure 5 ijms-23-08035-f005:**
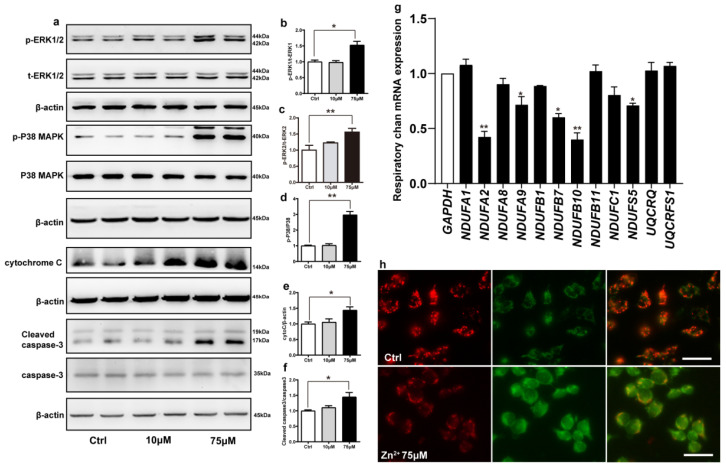
Zn^2+^ activates the ERK/P38 signaling pathway and causes mitochondrial damage in HEK293 α-synuclein-DsRed cells. (**a**–**e**) HEK293 α-synuclein-DsRed cells were treated with 10 or 75 μM ZnSO_4_ for 12 h. The p-ERK1 (**a**,**b**), p-ERK2 (**a**,**c**), p-p38 (**a**,**d**), cleaved caspase 3 (**a**,**f**) levels and the total protein levels of ERK1/2, p38, caspase 3 and expression levels of cytochrome C (**a**,**e**) were determined via Western blotting with β-actin as an internal control ((**a**), representative pictures; (**b**–**f**), quantification). (**g**) The mRNA expression levels of mitochondrial respiratory chain complex I and III subunits were determined via qRT-PCR using GAPDH as the internal controls. (**h**) JC-1 staining was used to detect the changes in mitochondrial membrane potential after HEK293 α-synuclein-DsRed cells were treated with 75 μM ZnSO4 for 12 h. Color dye changes reversibly from red to green as mitochondrial membranes become depolarized. All data from at least three independent experiments are represented as the means ± SEM. * *p* < 0.05, ** *p* < 0.01 with respect to the control group.

**Figure 6 ijms-23-08035-f006:**
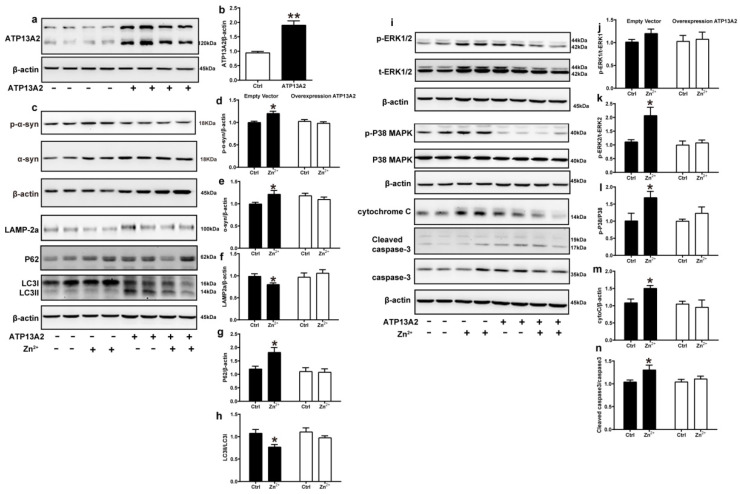
Overexpression of ATP13A2 inhibits zinc-induced autophagy–lysosome and ERK/P38 signaling pathways. The ATP13A2-pcDNA3.1-HA plasmid (ATP13A2 overexpression group) and the pcDNA3.1-HA empty vector (empty vector group) were transiently transfected into HEK293 α-synuclein-DsRed cells, and the ATP13A2 overexpression group and the empty vector group were treated or untreated with 75 μM ZnSO4 for 12 h. (**a**,**b**) ATP13A2 levels were detected through Western blotting and quantitative analyses (a, representative pictures; b, quantification). Autophagy–lysosome pathway (c, representative pictures; d-h quantification): the expression levels of p-α-synuclein (**c**,**d**), α-synuclein (**c**,**e**), LAMP-2a (**c**,**f**), P62 (**c**,**g**), LC3 I and LC3 II ((**c**,**h**), quantification of ratio of LC3II to LC3I) were determined using Western blot analysis with β-actin as an internal control. Apoptosis pathway ((**i**), representative pictures; j-n, quantification): the p-ERK1 (**i**,**j**), p-ERK2 (**i**,**k**), p-p38 (**i**,**l**), cleaved caspase 3 (**i**,**n**) levels and the total protein levels of ERK1/2, p38, caspase 3 and expression levels of cytochrome C (**i**,**m**) were determined via Western blotting with β-actin as an internal control. Data are represented as the means ± SEM of at least three independent experiments. * *p* < 0.05, ** *p* < 0.01 with respect to the control group.

**Figure 7 ijms-23-08035-f007:**
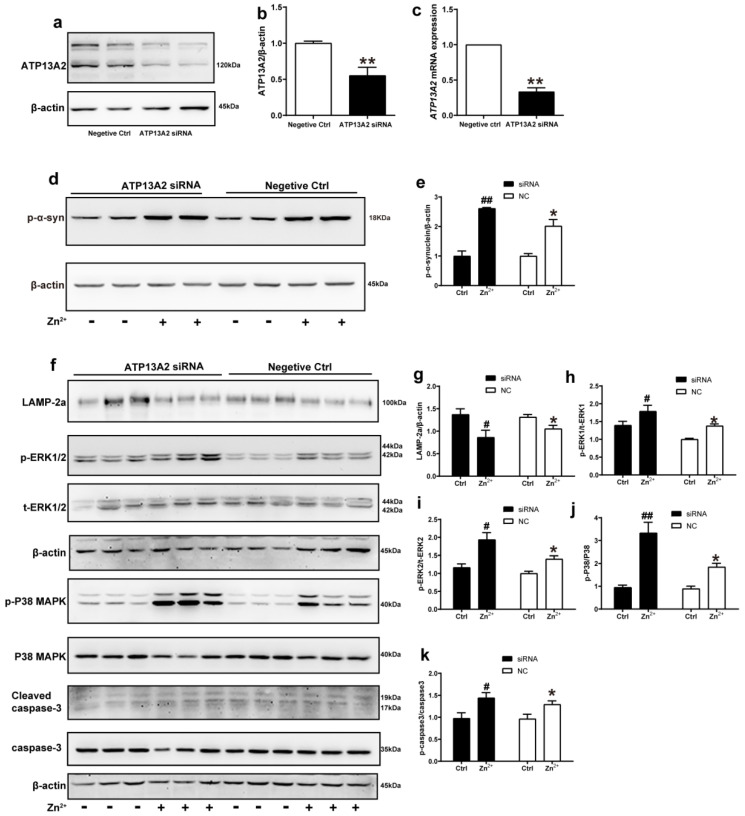
ATP13A2 siRNA promotes zinc-induced activation of the ERK/P38 signaling pathway. The ATP13A2 siRNA and the negative control were transiently transfected into HEK293 α-synuclein-DsRed cells, and then the cells were treated or untreated with 75 μM ZnSO4 for 12 h. (**a**,**b**) ATP13A2 levels were detected using Western blotting and quantitative analyses ((**a**), representative pictures; (**b**), quantification). (**c**) The mRNA expression levels of ATP13A2 were determined via qRT-PCR using GAPDH as an internal control ((**d**,**f**) representative pictures; (**e**,**g**–**k**) quantification). The expression levels of p-α-synuclein (**d**,**e**) and LAMP-2a (**f**,**g**), the phosphorylation and total protein levels of ERK1/2 (**f**,**h**,**i**), p38 (**f**,**j**) and caspase 3 (**f**,**k**) were determined using Western blotting with β-actin as an internal control. Data are represented as the means ± SEM of at least three independent experiments. * *p* < 0.05, ** *p* < 0.01 with respect to the control in the negative control group; # *p* < 0.05, ## *p* < 0.01 compared to the control in the ATP13A2 silencing group.

**Table 1 ijms-23-08035-t001:** List of reagents.

Reagents	Code Number	Company
Alexa Fluor^®^ 488-conjugated donkey anti-rabbit IgG	A-21206	Thermo Fisher Scientific, Waltham, MA, USA
Alexa Fluor^®^ 594-conjugated donkey anti-mouse IgG	A-21203	Thermo Fisher Scientific
Biotinylated goat anti-mouse IgG	E043301-2	Dako, Carpinteria, CA, USA
CellTiter 96^®^ AQueous One Solution Cell Proliferation Assay (MTS)	G3582	Promega, Madison, WI, USA
eBioscience™ JC-1DyeMitoMemPotentialDye	65-0851-38	Thermo Fisher Scientific, (Invitrogen), Waltham, MA, USA
GoScript Reverse Transcription System	A5001	Promega
GoTaq qPCR Master Mix	A6001	Promega
N-(6-methoxy-8-quinolyl)-p-toluenesulfonamide (TSQ)	M688	Thermo Fisher Scientific (Invitrogen)
Peroxidase AffiniPure Goat Anti-Mouse IgG (H + L)	115-035-003	Jackson Immuno Research,West Grove PA, USA
Peroxidase AffiniPure Goat Anti-Rabbit IgG (H + L)	111-035-003	Jackson Immuno Research
Phosphatase inhibitor cocktail	07574-61	Nacalai Tesque, Kyoto, Japan
Protease inhibitor cocktail	03969-21	Nacalai Tesque
ZnSO_4_·7H_2_O	221376	Sigma
3,3′-Diaminobenzidine (DAB)	D8001	Sigma, Burlington, MA, USA

**Table 2 ijms-23-08035-t002:** List of primary antibodies.

Antibody	Resource	Code Number	Company
ATP13A2	rabbit	NB110-41486	Novus, Englewood, CO, USA
BAX	rabbit	2772	Cell Signaling Technology, Danvers, MA, USA
BCL-2	rabbit	Ab196495	abcam, Cambridge, UK
Caspase-3	rabbit	9662	Cell Signaling Technology
Caspase-3 (cleaved)	rabbit	9664	Cell Signaling Technology
Cyt C	rabbit	11940	Cell Signaling Technology
LAMP-1	rabbit	ab24170	abcam
LAMP-2a	rabbit	Ab18528	abcam
LC3 Ⅰ/Ⅱ	mouse	4108	Cell Signaling Technology
P38/MAPK	rabbit	9212	Cell Signaling Technology
P38/MAPK (Phospho-Tyr182)	rabbit	4511	Cell Signaling Technology
P62	rabbit	8025	Cell Signaling Technology
PARK9 (ATP13A2)	rabbit	5879	Cell Signaling Technology
Phospho-p44/42MAP (ERK1/2) Thr202/Tyr204	rabbit	9101	Cell Signaling Technology
p44/42 MAPK (ER1/2)	rabbit	4695	Cell Signaling Technology
Tyrosine Hydroxylase	rabbit	3443922	Millipore, Burlington, MA, USA
α-Synuclein (phospho S129)	rabbit	Ab51253	abcam
α-synuclein (211)	mouse	sc-12767	SANTA, Dallas, TX, USA
β-actin	mouse	A1978	Sigma, Burlington, MA, USA

**Table 3 ijms-23-08035-t003:** Primer sequences.

Gene	Forward	Reverse
*GAPDH*	GAATGGGCAGCCGTTAGGAA	AAAAGCATCACCCGGAGGAG
*ATP13A2*	TTCCTGGCAGCTCTTCACTG	CTTCTGTCCGACACTCACCG
*NDUFA1*	GAAGCCAGGTCACCTTTCAA	TCCTGGAATCAACAAGCACA
*NDUFA2*	CCCGACCTACCCATCCTAAT	GGCCAAATGCTGAAGAGAGA
*NDUFA8*	AGCTGGGAGAACAACGACAC	GACCTGTGAGTTTGCCCAAT
*NDUFA9*	TCACGTTCTGCCATTACTGC	ATCCCACTGACTGAGGAACG
*NDUFB1*	ATGATTTGCTGGCGTCACCC	TGGTCCCGCACAATCTGAAG
*NDUFB7*	CTCAAGTGCAAGCGTGACAG	CCTTCATGCGCATCACATAG
*NDUFB10*	CAGCACGCAAAGAACAGGTA	TCCCTCTTCCACTGCATTTC
*NDUFB11*	CCCAAATGTCACCGATTTCT	AACCCTCTTTGCCTCCAGTT
*NDUFC1*	GAGTTGACGGACCGACCTTA	ATGGGGAGTTCGAGTCACAA
*NDUFS5*	CGGTTATACTCGGGCAGAGA	TATCAGCTTATCCCGCTGCT
*UQCRQ*	GCATTCGGGAGTCTTTCTTTC	TGCGTTGCTCATTTGTCATT
*UQCRFS1*	CCTCAATGTCCCTGCTTCTG	GCCTCGCTGCTTTCTCTTG

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
