# Peer review of "ATP13A2 Declines Zinc-Induced Accumulation of α-Synuclein in a Parkinson’s Disease Model"

_ijms, 2022, doi:10.3390/ijms23148035_

Round 1

Reviewer 1 Report

This paper has studied the role of ATP13A2 in high Zn2+ condition using α-synuclein overexpressed cells and mice. The authors show that the expression of ATP13A2 was increased in high Zn2+-treated α-synuclein overexpression model. Moreover, high Zn2+-induced pathologies were regulated by dependent of ATP13A2 expression. Although the authors provided clear results for the role of ATP13A2 in Zn2+ homeostasis, I would suggest authors to address several issues prior to its publication.

-          In all in vitro and in vivo experiments, the results should be tested in WT cells and mice (without α-synuclein overexpression) because α-synuclein overexpression could affect the expression of ATP13A2, TH, LAMP2a, P62, LC3 and etc. This allows us to understand the role of high Zn2+ condition depends on α-synuclein levels.

-          Page 5, line 148, in Fig. 1f, g and h, the authors investigate the expression of ATP13A2 only in high Zn2+ condition instead of α-synuclein levels. This paragraph would be changed to “Zn2+-affected the expression of ATP13A2” rather than “Zn2+ affected α-synuclein aggregation”.

-          For more experimental rigor, it would be better to present the result from co-staining with TH for ATP13A2 detection in SNc in Figure h

-          In Fig. 2f-j, to detect the pathological α-synuclein by antibody against pS129-α-synuclein, the authors should assess the protein levels in Triton X100-insoluble fraction rather than total cell lysates.

-          In Fig. 6, the quantitative analysis should be tested by 2-way ANOVA analysis with multiple comparison to compare the effect of ATP13A2 overexpression.

Minor comments,

1)      Please check any typos, grammar errors and synchronize the word (i,e. a-synuclein vs α-synuclein vs alpha-synuclein / ATP13A2 vs ATP13a2 / Dsred vs DsRed / Sirna vs siRNA)

2)      A lot of sentences are needed to insert references in sections of Introduction, Results and Discussion. Please check again.

3)      It would be better to arrange the figure format according to the manuscript (Figure 2d > 2b, 3e > 3d)

Reviewer 2 Report

In this manuscript, the authors present a study of the role of the transporter ATP13A2 in the pathway of the zinc homeostasis and the a-synuclein accumulation. With that propose they employ an α-synuclein-GFP transgenic mice model and HEK293 α-synuclein-DsRed cell line model.

The manuscript is well written and is easy to read. I would like to recommend the publication of this manuscript and I have only a number of questions that I think would be interesting:

1)  Why was it decided to use the ZnSO4 concentrations of 20 ppm and 200 ppm in water for the mice and not more? I understand that the 20 ppm is of interest because it is an amount that allows observing certain changes, but in a slight way. But why 200 ppm? Why weren't intermediate or higher amounts of zinc salt used to see if there was a linear relationship between the amount of zinc salt added and the effects on the specimens?

2)  Regarding the treatment of the chosen cell model with zinc salt, why were these two concentrations (10 µM and 75 µM) of zinc solution chosen?

3)  Do the authors know the equivalence of zinc concentration to which the neuronal cells of mice are exposed to the two different doses of zinc salt supplied with the water? I mean, does the 200 ppm dose given to the mice maintain any equivalence with the 75 µM dose given to the model cells?

Round 2

Reviewer 1 Report

Although the revised figure was not improved during this revision, the authors put into many attempts to add experimental results in the revised manuscript. Moreover, the authors have addressed most of my comments. This reviewer recommend to accept the revised manuscript for publication.